# Description of the China global Merged Surface Temperature version 2.0

Wenbin Sun[1,#], Yang Yang[1,#], Liya Chao[1,#], Wenjie Dong[1,#], Boyin Huang[2], Phil Jones[3],

Qingxiang Li[1,#]

[1] *School of Atmospheric Sciences and Key Laboratory of Tropical Atmosphere-Ocean System, Sun Yat-sen University, Ministry of Education, Guangzhou, China*

[2] *National Centers of Environmental Information, NOAA, Asheville, USA*

[3] *Climate Research Unit, University of East Anglia, Norwich, UK*

[#] *Current address: Southern Laboratory of Ocean Science and Engineering (Guangdong Zhuhai), Zhuhai, China*

*Corresponding to*: Qingxiang Li (liqingx5@mail.sysu.edu.cn)

**Abstract.** Global surface temperature observational datasets are the basis of global warming studies. In the context of increasing global warming and frequent extreme events, it is essential to improve the coverage and reduce the uncertainty of global surface temperature datasets. The China global Merged Surface Temperature Interim version (CMST-Interim) is updated to CMST 2.0 in this study. The previous CMST datasets were created by merging the China global Land Surface Air Temperature (C-LSAT) with sea surface temperature (SST) data from the Extended Reconstructed Sea Surface Temperature version 5 (ERSSTv5). The CMST2.0 contains three variants: CMST2.0-Nrec (without reconstruction), CMST2.0-Imax, and CMST2.0-Imin (According to their reconstruction area of the air temperature over the sea ice surface in the Arctic region). The reconstructed datasets significantly improve data coverage, whereas CMST2.0-Imax and CMST2.0-Imin have improved coverage in the Northern Hemisphere, up to more than 95%, and thus increased the long-term trends at global, hemispheric, and regional scales from 1850 to 2020. Compared to CMST-Interim, CMST2.0-Imax and CMST2.0-Imin show a high spatial coverage extended to the high latitudes and are more consistent with a reference of multi-dataset averages in the polar regions. The CMST2.0 datasets presented here are publicly available at the website of figshare, https://doi.org/10.6084/m9.figshare.16929427.v4 (Sun and Li, 2021a)and the CLSAT2.0 datasets can be downloaded at https://doi.org/10.6084/m9.figshare.16968334.v4 (Sun and Li, 2021b).

## 1. Introduction

Global Surface Temperature (GST) is a key meteorological factor in characterizing climate change and has been widely used for climate change detection and assessment (IPCC, 2013; 2021). GST consists of global Land Surface Air Temperature (LSAT), which is the 2-m air temperature observed by land weather stations, and Sea Surface Temperature (SST) observed by ships, buoys and Argos. However, there are large uncertainties in the temperature data observed by weather stations, ships, buoys and Argos in long-term observations, including uncertainties due to uneven spatial and temporal distribution of sampling (Jones et al., 1997; Brohan et al., 2006) and





uncertainties due to stations, environment and instrumentation changes (Parker et al., 1994; Parker,
2006; Trewin, 2012; Kent et al., 2017; Menne et al., 2018; Xu et al., 2018). Nevertheless, several
countries and research teams have applied different homogenization methods to generate a series of
representative homogenized global land-sea surface temperature gridded datasets, including the Met
Office Hadley Centre/Climatic Research Unit Global Gridded Monthly Temperature (HadCRUT)
(Morice et al., 2012), Goddard Institute for Space Studies Surface Temperature (GISTEMP)
(Hansen et al., 2010; Lenssen et al., 2019), NOAA's NOAA Global Temperature
(NOAAGlobalTemp) (Vose et al., 2012; Zhang et al., 2019; Huang et al., 2020), and Berkeley Earth
(BE) (Rohde et al., 2013a; Rohde and Hausfather, 2020), which serve as benchmark data for
monitoring and detecting GST changes and related studies.
However, there are still uncertainties in these datasets, including those due to insufficient
coverage, especially at high altitudes and in the polar regions. The Artic has high climate sensitivity
(Lu and Cai, 2009, 2010; Yamanouchi, 2011; Dai et al., 2019; Xiao et al., 2020; Latonin et al., 2021),
the absence of data for this region would lead to a cold bias in the estimated global mean surface
temperature (GMST). How to improve this deficiency is an issue that must be addressed to optimize
and improve the observations. Since IPCC AR5 (2013), all of the above datasets have been updated
and reconstructed in the data default region (IPCC, 2021). For example, Cowtan and Way (2014)
used kriging and hybrid methods to fill in the HadCRUT4 data gap areas, extending the data to polar
regions. GISSTEMP v4 utilized spatial interpolation methods to fill in the default data within the
appropriate distances (1200km) (Lenssen et al., 2019). NOAA/NCEI used spatial smoothing and
empirical orthogonal remote correlations (EOTs) to reconstruct the data default areas, generating
100-member GHCN ensemble data and 1000-member ERSST ensemble data, respectively, which
were combined into the NOAAglobalTemp-Interim dataset (Vose et al., 2021). HadCRUT team
infilled HadCRUT5 using the Gaussian process method (Morice et al., 2021). Kadow et al. (2020)
used artificial intelligence (AI) in combination with numerical climate model data to fill the
observation gaps in HadCRUT4. Berkeley Earth used kriging-based spatial interpolation to fill in
the terrestrial default data (Rohde et al., 2013a; Rohde et al., 2013b; Rohde and Hausfather, 2020).
Interpolation and reconstruction for high latitudes reduce the error in the estimate of GMST.
Compared to 0.61 (0.55-0.67) °C in IPCC AR5, GST warming estimated with reconstructed datasets
in AR6 from 1850-1800 to 1986-2005 is 0.69 (0.54-0.79) °C, which increased 0.08 (- 0.01 to
0.12) °C (IPCC, 2021).
China global Merged Surface Temperature (China-MST or CMST) is a new global surface
temperature dataset developed by the team at Sun Yat-sen University, which was merged by China
global Land Surface Air Temperature (China-LSAT or C-LAST) (Xu et al., 2018; Yun et al., 2019;
Li et al., 2020; Li et al., 2021) as the terrestrial component and ERSSTv5 (Extended Reconstructed
Sea Surface Temperature version 5) (Huang et al., 2017) as the ocean component. It is generally
consistent with other global datasets in terms of GST trends and uncertainty levels since 1880 (Li et
al., 2020). Compared with other datasets, the station coverage of C-LSAT has been significantly
improved, especially for Asia (Xu et al., 2018), and more ISTI station data have been added in C-
LSAT 2.0 (Li et al., 2021; Thorne et al., 2011). In addition, C-LSAT adopted a homogenization
scheme for temperature series that is different from datasets such as the Global Historical
Climatology Network version 4 (GHCNm v4) (Menne et al., 2018). Further, Sun et al. (2021) trained
EOTs modes with "state-of-the-art" ERA5 reanalysis data to extract the spatial distribution of LSAT.
They then used a similar low- and high-frequency reconstruction method of Huang et al. (2020)





with different parameter schemes, combined with the observation constraint method, to fill the data
default region of C-LSAT2.0 and released the new reconstructed dataset C-LSAT2.0 ensemble and
the global surface temperature dataset CMST-Interim. Compared with the original CMST, CMST-
Interim significantly improves the coverage of GST, and the GST warming estimated by CMST-
Interim is more significant, with the warming trend since the 1900s increasing from $0.085 \pm 0.004°C$
$(10 \text{ yr})^{-1}$ to $0.089 \pm 0.004°C$ $(10 \text{ yr})^{-1}$. In the current CMST-Interim (Sun et al., 2021) and its earlier
version (Yun et al., 2019), we still fully adopted the setting from ERSSTv5, which treats the sea ice
region in the Arctic as the sea surface temperature below the sea ice and assigns a default value (-
1.8°C), which makes it still a gap in the polar region. In contrast, polar regions are susceptible to
climate forcing, with the Arctic warming more than twice the global average in recent decades
(Goosse et al., 2018). The lack of data from CMST-Interim in polar regions may result in a slight
underestimation of its estimated global warming trend. Furthermore, CMST-Interim does not
systematically assess the reconstruction uncertainty of LSAT, resulting in an incomplete estimate of
global surface temperature uncertainty (Li et al., 2021). Although C-LSAT 2.0 ensemble satisfied
the criterion of the recently released the 6[th] assessment report of IPCC, the CMST -Interim does not
appear in the core assessment GMST series due to its insufficient data coverage in the Arctic region
(Gulev et al, 2021).
To address the above issue and improve coverage of CMST in the Arctic, we further reconstruct
and supplement the Arctic data default region in the dataset using a combination of statistical
interpolation and high- and low-frequency reconstruction to develop the reconstructed CMST2.0
dataset and assess its uncertainty. Section 2 introduces the update of terrestrial and oceanic datasets,
section 3 presents the reconstruction and uncertainty analysis of CMST2.0, section 4 introduces the
composition of C-LSAT2.0 and CMST2.0, section 5 analyzes the GMST series of CMST2.0, section
6 is the comparison of CMST2.0 dataset with other datasets, section 7 provides the summary and
outlook, and section 8 is data availability.
**2.   Updates of the land and ocean datasets**
**2.1 Data sources and initial processing for C-LSAT2.0**
The initial version of the C-LSAT dataset was C-LSAT1.0. The C-LSAT1.0 site dataset
collected and integrated 14 LSAT datasets, including three global data sources (CRUTEM4, GHCN-
V3, and BEST), three regional data sources, and eight national situ data sources (Xu et al., 2018).
The current latest version is C-LSAT 2.0 (Li et al., 2021; Sun et al., 2021).
C-LSAT 2.0 used in this study is an update of C-LSAT 1.3. Compared to C-LSAT 1.3 from
1900 to 2017, version 2.0 extend to 1850-2020, and there is a significant increase in the amount of
in situ data for the period 2013-2017 (Figure 1), with the increased situ data from CLIMAT from
WMO's Global Telecommunication System (GTS) and Global Surface Daily Summary (GSOD)
(https://www.ncei.noaa.gov/data/global-summary-of-the-day/archive/; last access: November 2021)
and is homogenized using the same method as Xu et al. (2018). In addition, we have updated the
data in C-LSAT2.0 for 2013-2019, which adds the number of situ data in Africa, North America and
other regions in this study. The C-LSAT 2.0 dataset includes three temperature elements: monthly
mean temperature, maximum temperature, and minimum temperature, and its time range for the
three elements is January 1850 - December 2020.

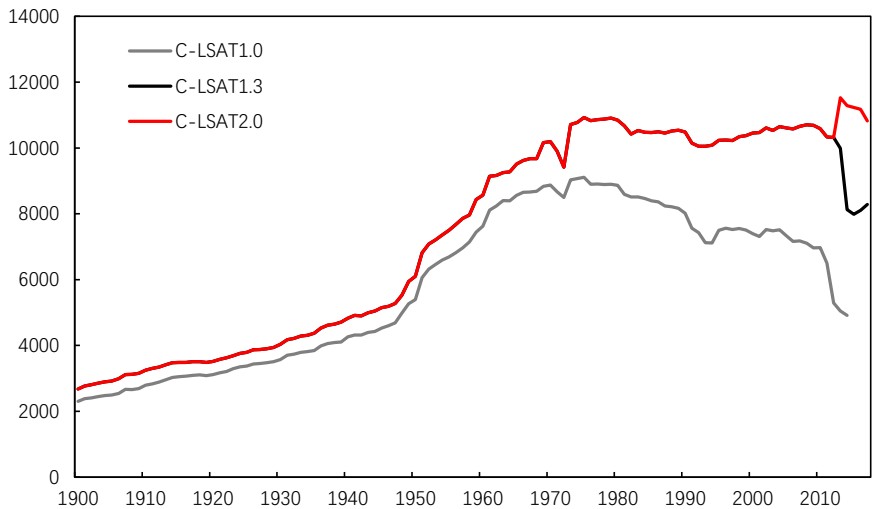


Figure 1 Comparison of C-LSAT 1.3 and C-LSAT 2.0 site counts from 1900 to 2017
**2.2  Sea surface temperature**
CMST1.0 (Yun et al., 2019) and CMST-Interim (Sun et al., 2021) use ERSSTv5 as the ocean
component (Huang et al., 2017). ERSSTv5 starts from 1854, and we extend ERSSTv5 (1854-present)
to 1850 using 1850-1853 SST anomalies (relative to 1961-1990 average) from ICOADS Release
3.0 (Freeman et al., 2017) and integrated into a global SST anomaly dataset for January 1850 -
December 2020. In the above integrated SST dataset, the SST is still set to a constant value of -
1.8°C for areas with >90% sea ice coverage as ERSSTv5. In addition, some areas in the high
latitudes of the Southern Hemisphere (non-sea ice) are marked as missing values due to the lack of
observations.
**2.3  Sea ice surface air temperature**
The common air temperature observation for the Arctic region is The International Arctic Buoy
Program (IABP) (http://research.jisao.washington.edu/data_sets/iabppoles/; last access: October
2021), which contains oceanographic and meteorological observations for the Pacific Arctic, but it
only has sea ice data from 1979 to the present, while the climate state of CMST is 1969-1998, the
time length of IABP does not support us to estimate and reconstruct the temperature anomaly of the
Arctic region in the CMST dataset, so we use the Inverse Distance Weighted (IDW) extrapolation
(site data) and EOT interpolation (gridding) methods to fill the default grid of the polar region
(Cowtan and Way, 2014; Lenssen et al., 2019; Rohde and Hausfather, 2020; Vose et al., 2021).
**3.  CMST2.0 reconstruction and uncertainty analysis**
**3.1 CMST and its brief reconstruction history**
CMST 1.0 consists of C-LSAT 1.3 (1900-2017) as the terrestrial component and ERSSTv5 as
the ocean component. The latest version without reconstruction is CMST2.0-Nrec in this study,
which composes of C-LSAT2.0 and ERSSTv5. Compared to CMST1.0 from 1900-2017, CMST2.0-
Nrec has been updated and expanded to 1850-2020. The original reconstructed version of CMST is
the Chinese global merged surface temperature reconstruction dataset CMST-Interim, which is a
merge of the reconstructed C-LSAT2.0 and ERSSTv5, where the reconstructed C-LSAT2.0 is an
ensemble reconstruction dataset upgraded from C-LSAT2.0 (Li et al., 2021) with 756 ensemble





members identified based on EOT and smoothing (Sun et al., 2021). Considering that there are much
missing data due to sea ice coverage at high latitudes in the Northern Hemisphere in CMST, the
IDW extrapolation method is proposed to infill the missing data in some key sites, then EOT
interpolation method is used to reconstruct all the grid boxes over the sea-ice-covered region in this
paper. Considering the effect of interannual variability of sea ice in the Arctic, 65ºN-90ºN and 80ºN-
90ºN are taken as the assumed land components for ensemble reconstruction with C-LSAT 2.0,
respectively, using the maximum sea ice area and minimum sea ice area since satellite observations
are available as reference, then the ERSSTv5 ensemble reconstruction dataset is merged to generate
CMST 2.0-Imax and CMST 2.0-Imin datasets.
**3.2 Reconstruction of terrestrial and marine components**
**3.2.1 Reconstruction of the terrestrial component**
We follow the reconstruction method of CMST-Interim (Sun et al., 2021) and divide the C-
LSAT 2.0 dataset into two parts, high- and low-frequency components, for reconstruction, then sum
them to obtain the reconstructed LSAT data (Figure 2). The low-frequency component is a running
average over time and space to characterize the large-scale features of LSAT anomalies in time and
space. First, a 25° x 25° spatial running average is performed, and then the annual average of LSAT
anomalies is calculated for at least two months of the year. Then, a 15-year median filter is used for
the annual average LSAT, followed by a 15° x 25° spatial sliding average, a 9-point binomial spatial
filter, and a 3-point binomial temporal filter for latitude and longitude, respectively, to fill in the
default data. Finally, a 15° x 25° spatial running average is applied to latitude and longitude
respectively to smooth the spatial distribution of the LSAT. The high-frequency component is the
difference between the original data and the low-frequency component, characterizing the local
variation of LSAT. We train the EOTs modes using the ERA5 reanalysis dataset (Hersbach et al.,
2020) (https://cds.climate.copernicus.eu/; last access: July 2020) and localize it. Afterward, the
EOTs modes are used to fit the high-frequency data to obtain a full-coverage reconstruction of the
high-frequency component (Sun et al., 2021). The reconstructed land temperature data can be
obtained by summing the low-frequency and high-frequency components, and finally, the
reconstructed data are observationally constrained to remove the low-quality reconstructed data.

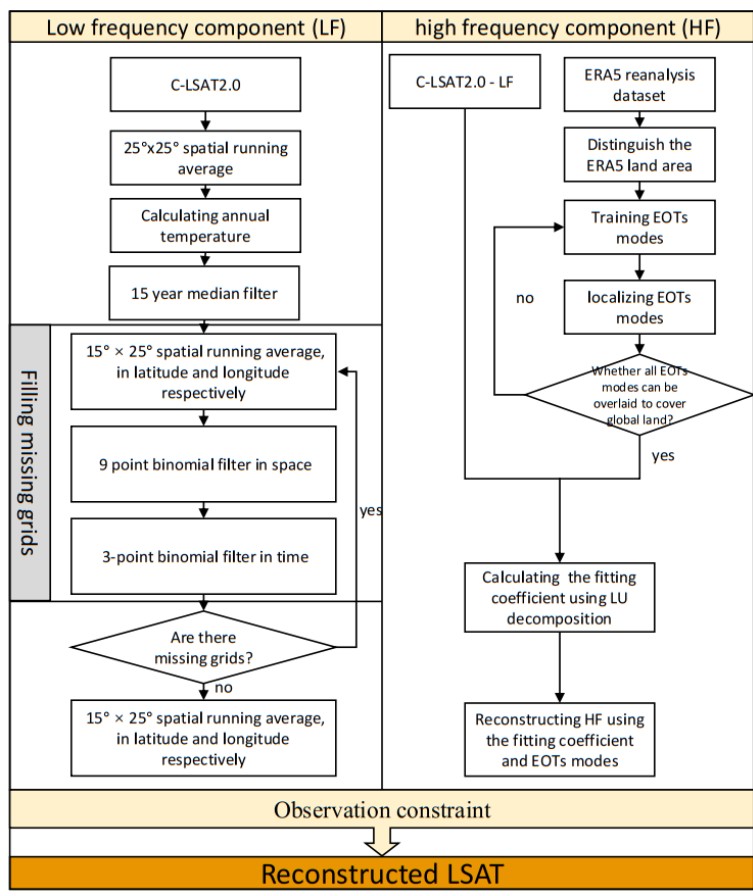

Figure 2 Schematic diagram of the LSAT reconstruction process

Reconstruction greatly improves the coverage of C-LSAT2.0. Figure 3 shows the comparison of land coverage before and after reconstruction. The land coverage of the reconstructed C-LSAT2.0 increases from the original 4.6% in 1850 to 29%, and the land coverage remains above 60% after 1913 and reaches the maximum land cover of about 80% in 1961, which last until 1990, after which it slightly decreases and remains at about 78%. After 2012 there is a decreasing trend to about 70%, where the land cover in 2019 is the lowest value of 66% for the period 2012-2020, this is related to the lower number of sites in the year.

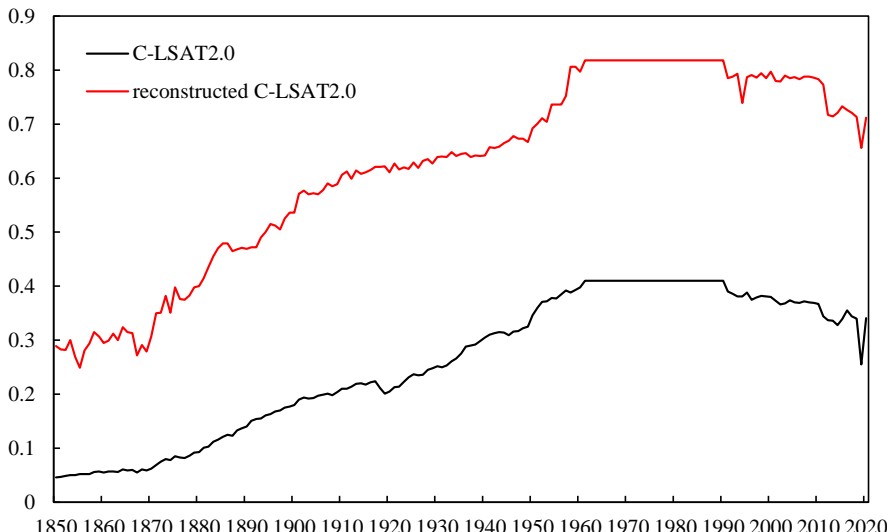

Figure 3 Coverage comparison of the terrestrial component before and after reconstruction

**3.2.2 Reconstruction of the ocean component**

We use ERSSTv5 data as the basis, which is a full-coverage, monthly reconstructed SST dataset based on observations from ships, buoys, and Argo (Huang et al., 2017). We fill the data during 1850-1853 with SST anomaly observed by ICOADS Release 3.0 (Freeman et al., 2017) to form a complete monthly SST anomaly dataset from 1850-2020 and then reconstruct it using the EOTs of Huang et al. (2017) to reduce the missing data.

**3.3 Reconstruction of Arctic ice surface temperature**

In CMST-Interim, when the Arctic is covered by sea ice, ERSSTv5 sets SST in the region with >90% sea ice coverage to a constant value (-1.8°C), making ST of CMST-Interim in the polar region the default value. It is worth noting that the Arctic is extremely sensitive to changes in climate forcing (polar amplification effect), so missing data in the polar regions in CMST-Interim may lead to an underestimation of the global warming trend (IPCC, 2021).

In order to solve this problem and improve the coverage of CMST in the Arctic, we improve the ST reconstruction method in the Arctic by expressing the ST of the Arctic in terms of the air temperature of ice surface (considering the similar physical properties of ice and land, the sea ice is considered as the land). The month with the largest extent of Arctic sea ice is March, and the month with the smallest extent is September. According to the National Snow and Ice Data Center, during 1980-2020, the year with the largest sea ice extent in March is 1983 and the year with the smallest sea ice extent in September is 2012, so we designed two experiments: 1) CMST2.0-Imax uses 2 m air temperature to represent the temperature within the 65ºN-90ºN region to simulate the ST of the Arctic sea ice-covered region in March 1983, which is the maximum sea ice extent. 2) CMST2.0-Imin uses 2 m air temperature to represent the temperature within the 80ºN-90ºN region to represent the ST in the Arctic sea ice-covered region at the time of September 2012, which is the minimum sea ice extent (Figure 4).



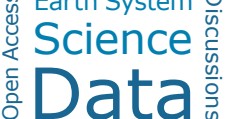

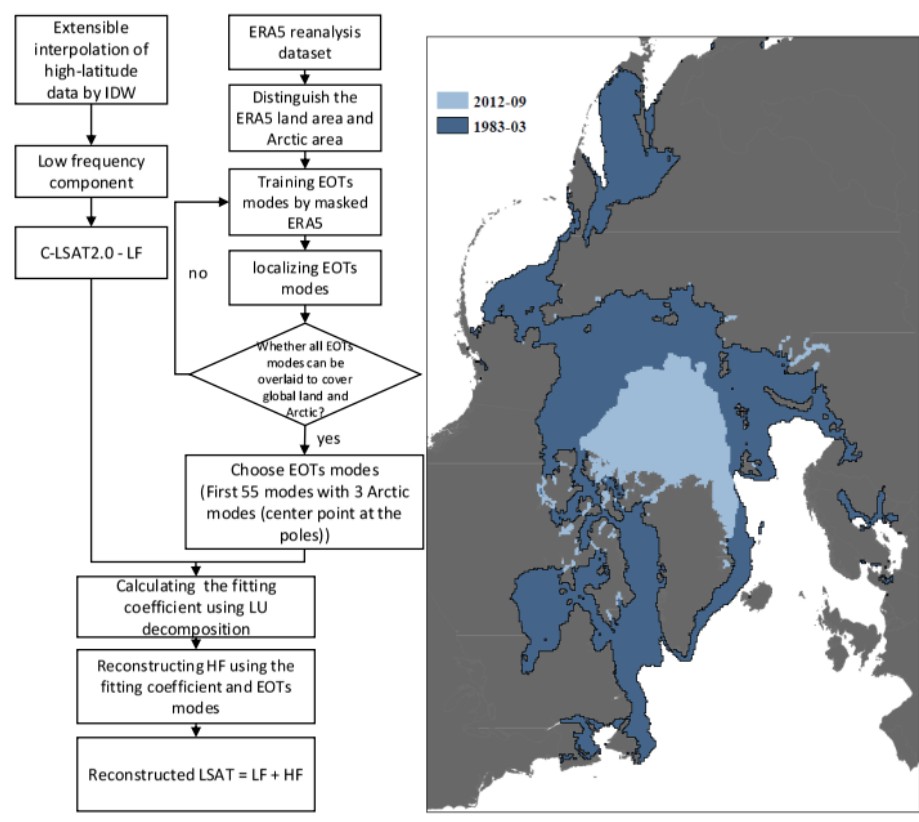

Figure 4 Reconstruction process of Arctic sea ice ST (left); comparison of maximum sea ice extent (sea ice extent in March 1983, shaded in dark blue) and minimum sea ice extent(sea ice extent in September 2012, shaded in light blue) distribution (right)

### 3.3.1 Maximum sea ice extent reconstruction CMST2.0-Imax

Due to the scarcity of observations in the Arctic and the fact that most observations were available after the 1980s, the observation period is very short. The data do not cover all the period of 1961-1990, which is the climatology of our dataset. Therefore the observations cannot be added to the C-LSAT 2.0 dataset. Due to this fact, we use the Inverse Distance Weighted method (IDW) (Cheng et al., 2020) to interpolate the data at lower latitudes to the Arctic (65°N-90°N) and then perform the high- and low-frequency reconstruction method based on the interpolated dataset. It is worth noting that we included the region of 65°N-90°N when training EOTs using the ERA5 reanalysis dataset. We selected the first 55 modes of the EOTs with three polar modes (the center point at the Arctic poles), for a total of 58 modes for reconstructing the high-frequency components (Figure 4). After that, the reconstructed C-LSAT is merged with ERSSTv5, where the merged ERSSTv5 covers only the region south of 65°N.

### 3.3.2 Minimum sea ice extent reconstruction CMST2.0-Imin

The reconstruction method of the terrestrial component in CMST2.0-Imin is consistent with CMST2.0-Imax, except that the merged process with ERSSTv5, in CMST2.0-Imin, the merged ERSSTv5 coverage is south of 80N. It is worth noting that the sea ice coverage range is 80°N-90°



N and the region of 65°N-80°N fill in SST in CMST2.0-Imin. However there are some grids in the
region of 65°N-80°N that are default values (caused by sea ice coverage) in ERSSTv5, so we use
the IDW method to fill these default grids.
Figure 5 shows the coverage comparison of CMST2.0-Nrec (without any land and ice air
temperature reconstruction), CMST-Interim, CMST2.0-Imax, and CMST2.0-Imin. Overall, there is
a significant improvement in the coverage of the reconstructed datasets compared to the original
dataset, CMST2.0-Nrec. Globally, the coverage of CMST2.0-Imax and CMST2.0-Imin
reconstructed for Arctic sea ice is consistently higher than CMST-Interim. CMST2.0-Imax and
CMST2.0-Imin have the highest global coverage, with >80% coverage after 1899. The global
coverage of CMST-Interim reached more than 80% after 1957. The comparative results for Northern
Hemisphere coverage are primarily consistent with the global, with CMST2.0-Imax and CMST2.0-
Imin having the greatest coverage, both reaching more than 90% after the 1880s, and CMST-Interim
reaching 80% coverage in 1901, but consistently below 90%. In terms of global and Northern
Hemisphere coverage, there are differences between CMST2.0-Imax, CMST2.0-Imin, and CMST-
Interim, but the differences are not significant. However, the coverage of CMST2.0-Imax and
CMST2.0-Imin differed significantly from CMST-Interim at high latitudes in the Northern
Hemisphere, where the coverage of CMST-Interim has been below 70% due to the existence of sea
ice, while CMST2.0-Imax and CMST2.0-Imin reache full coverage at high latitudes in the Northern
Hemisphere after 1983. There is no difference in the coverage of the three reconstructed datasets in
other regions (Southern Hemisphere, Southern Hemisphere mid-high and low latitudes) except for
the Northern Hemisphere and Northern Hemisphere high latitudes. The coverage of the
reconstructed dataset in the Southern Hemisphere has improved considerably, with a maximum
coverage of about 80%. The coverage of the reconstructed dataset in the high latitudes of the
Southern Hemisphere is relatively small, consistently below 50%, due to the scarcity of observations
in Antarctica.

Earth System
Science
Data

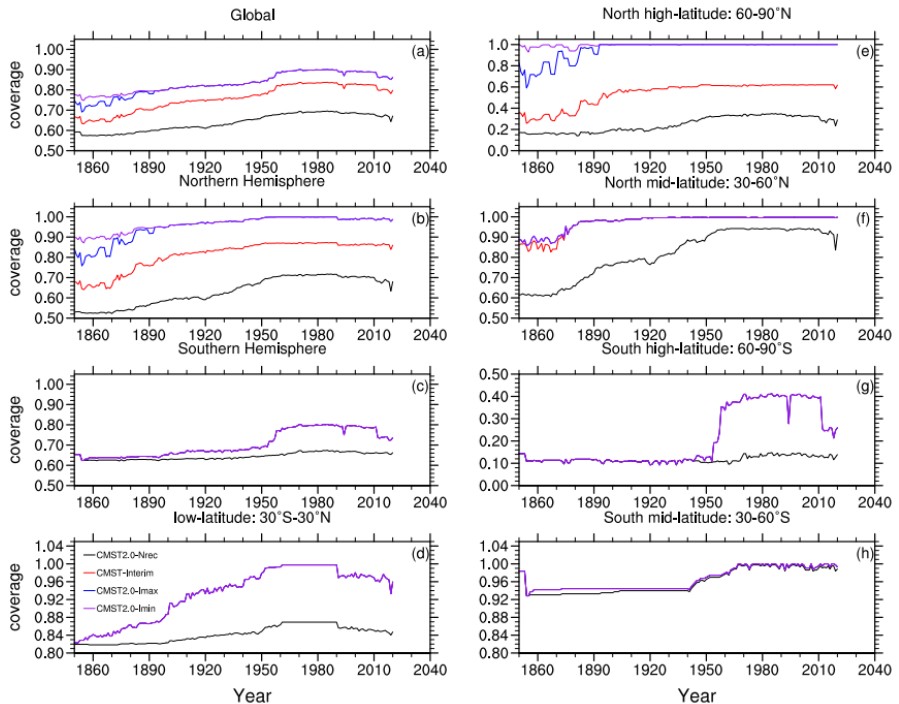

Figure 5 Coverage comparison of CMST2.0-Nrec, CMST-Interim, CMST2.0-Imax and CMST2.0-Imin

**3.4 Estimation of uncertainty in the reconstructed CMST2.0**

Uncertainties of the reconstructed CMST2.0 include both land and ocean uncertainties. The ocean uncertainty is the uncertainty of ERSSTv5. The land uncertainty is based on the reconstructed C-LSAT2.0 ensemble, which is divided into two parts: parameter uncertainty and reconstruction uncertainty. Since we reconstruct the temperature of the polar sea ice region in the way that we reconstruct the LSAT, we calculate the uncertainty of the 65°N-90°N (Imax) and 80°N-90°N (Imin) regions of CMST2.0-Imax and CMST2.0-Imin following the method of calculating the land uncertainty.

**3.4.1 Parameter uncertainty of C-LSAT2.0 ensemble**

In the reconstruction process, we choose different parameters to generate 756-member ensembles (Table 1), which are different for different combinations, so the parameter uncertainty represents the difference of parameter combinations. According to Huang et al. (2020), the parameter uncertainty (Up) is the regional average LSAT uncertainty, as follows:

$$U_p^2(t) = \frac{1}{M} \sum_{m=1}^{M} [A_m^g(t) - \overline{A^g}(t)]^2 \tag{1}$$

$$\overline{A^g} = \frac{1}{M} \sum_{m=1}^{M} A_m^g(t) \tag{2}$$

where M is the ensemble member, in this paper M=756; $A_m^g$ represents global LSAT of m-member

278 ensemble; $\overline{A^g}$ is the average of all ensembles; t represents temporal variations.

279 Table 1 Parameter settings used for reconstruction scenarios and the operational option.

| PARAMETER | OPERATIONAL OPTIONS | ALTERNATIVE OPTIONS |
|---|---|---|
| MINIMUM NUMBER OF MONTHS ANNUAL AVERAGE | 2 months | 1, 2, 3 months |
| LF FILTER PERIODS | 15 years | 10, 15, 20 years |
| MIN NUMBER OF YEARS FOR LF FILTER | 2 years | 1, 2, 3 years |
| EOTS TRAINING PERIODS AND SPATIAL SCALES | 1979-2018, Lx=4000, 3000, 2500, Ly=2500 | 1979-2018, Lx=3000,2000,1500, Ly=1500; 1979-2018, Lx=5000,4000,3500, Ly=3500; Lx=4000,3000,2500, Ly=2500; 1979-2008, Lx=4000,3000,2500, Ly=2500; 1989-2018, Lx=4000,3000,2500, Ly=2500; even year, Lx=4000, 3000, 2500, Ly=2500; odd year, Lx=4000, 3000, 2500, Ly=2500; |
| EOTS ACCEPTANCE CRITERION | 0.2 | 0.10, 0.15, 0.20, 0.25 |

280   Parameter uncertainties for the reconstructed C-LSAT2.0 ensemble, reconstructed C-
281 LSAT2.0+Imax (65°N-90°N) and reconstructed C-LSAT2.0+ Imin (80°N-90°N) show similar
282 variations. The parameter uncertainties decreases over time, as does its interannual variability. The
283 parameter uncertainties stabilizes below 0.05 during 1876-2016 (Figure 7). However, the parameter
284 uncertainty is higher in 2018-2020 compared to the previous years. This is due to the lower coverage
285 in this period compared to the last years, which is more sensitive to the parameter settings.

286 **3.4.2 Reconstruction uncertainty of C-LSAT2.0 ensembles**

287   In the reconstruction process, we smooth the observations when calculating the low-frequency
288 component to filter out the short-term and local signals to obtain the large-scale characteristics of
289 the LSAT anomaly, after which the high-frequency component is used to fit the local distribution of
290 LSAT using the EOTs spatial modes and the available observations. Our purpose of using EOTs is
291 to obtain the spatial distribution of the LSAT anomaly, filter out the errors in the observations, and
292 thus estimate the distribution of the LSAT anomaly from limited observations. However, the spatial
293 pattern of EOTs also smoothes out the local temperature and ignores some local information, thus
294 deviating from the observations. Therefore, according to Huang et al. (2016), we define the residual
295 between the ideal observations and the reconstructed values using EOTs as the reconstruction
296 uncertainty:

$$U_r^2(t) = \frac{1}{M} \sum_{m=1}^{M} [R_m^g(t) - D(t)]^2 \qquad (3)$$

297 where $D(t)$ represents the ideal observation and $R_m^g(t)$ is the reconstructed data obtained using
298 the high- and low-frequency reconstruction method based on $D(t)$.

299   The reconstruction uncertainty represents the differences between the ideal observations and
300 the reconstructions. We choose two full-coverage CMIP6 models to represent the ideal observations
301 to assess the deviation of the reconstructed values from the original values, which is due to missing
302 information caused by the smoothing of local temperatures by EOTs. The C-LSAT 2.0 ensemble
303 dataset covers the period 1850-2020, while the CMIP6 model historical experimental data are only

available up to 2014, so we use model data from the SSP370 scenario (taking into account minor
differences in the short term for any scenarios) to complement that of 2015-2020.
The two models we selected are BCC-CSM2-MR and GFDL-ESM4. BCC-CSM2-MR is a new
version of the climate system model developed by the National Climate Center of China with
improved parameterization and physical parameterization results. GFDL-ESM4 is an Earth system
model developed by the GFDL model of NOAA's Geophysical Fluid Dynamics Laboratory. Both
models have a resolution of 1.125 × 1.125, and we descale both to 5 × 5 to calculate the temperature
anomaly (1961-1990 climatology), after which the data from both models are reconstructed
according to the high- and low-frequency reconstruction method.
Figure 6 shows the reconstruction uncertainties calculated using BCC-CSM2-MR and GFDL-
ESM4. In general, the reconstruction uncertainties are relatively stable, do not increase over time.
The reconstruction uncertainties of reconstructed C-LSAT2.0+Imax and reconstructed C-LSAT2.0+
Imin are larger than that of reconstructed C-LSAT2.0, and the interannual variation is also larger.
The interannual variability of the uncertainty of BCC-CSM2-MR is slightly smaller than that of
GFDL-ESM4. In the following, we choose BCC-CSM2-MR as the reconstruction uncertainty to
discuss the uncertainty of the terrestrial component.

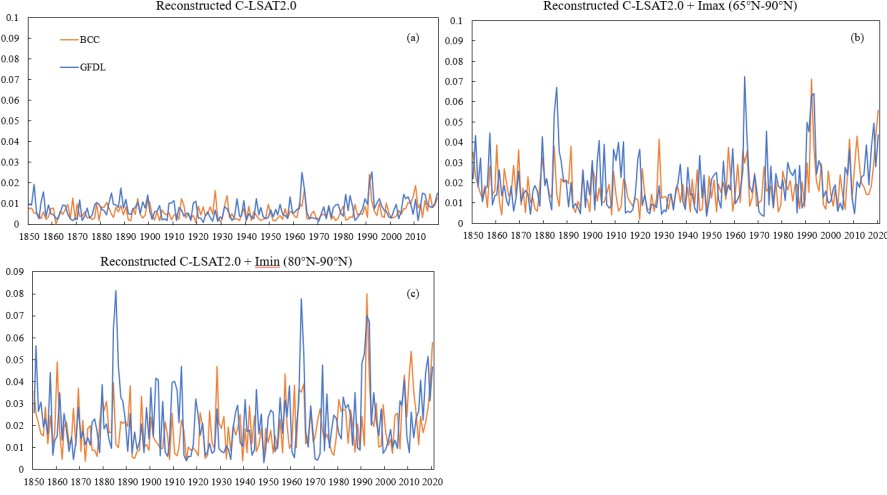


Figure 6    Reconstruction uncertainty of the reconstructed C-LSAT2.0 ensemble,
reconstructed C-LSAT2.0+Imax (65°N-90°N) and reconstructed C-LSAT2.0+ Imin (80°N-90°N)
calculated using BCC-CSM2-MR and GFDL-ESM4.

### 3.4.3 Total uncertainty of LSAT

The total uncertainty of the C-LSAT2.0 ensemble is the sum of the parameter uncertainty and
the reconstruction uncertainty:

$$U_l^2 = U_p^2 + U_r^2 \qquad\qquad (4)$$

Figure 7 shows the comparison of parameter uncertainty, reconstruction uncertainty and total
uncertainty of three C-LSAT2.0 ensemble datasets. The parameter uncertainties of the reconstructed
C-LSAT2.0 ensemble, reconstructed C-LSAT2.0+Imax (65 ° N-90 ° N) and reconstructed C-
LSAT2.0+ Imin (80°N-90°N) are much larger than the reconstruction uncertainties before 1950,
when the parameter uncertainties mainly determines the magnitude of total uncertainties. The
difference between the parameter uncertainties and the reconstruction uncertainties from 1950-2016
becomes small, and both determine the total uncertainties. The total uncertainties increase after 2017
due to the increase in parameter uncertainties (Figure 7a). The uncertainties of reconstructed C-
LSAT2.0+Imax and C-LSAT2.0+Imin vary similarly (Figure 7b&7c). The parameter uncertainties
of reconstructed C-LSAT2.0-Imax and C-LSAT2.0-Imin is larger than the reconstruction
uncertainties before 1880, when the total uncertainties is dependent on parameter uncertainties.
During 1880-1950, the magnitude and variation of the parameter uncertainties and the
reconstruction uncertainties are similar. After 1950, the parameter uncertainties decrease to less than
the reconstruction uncertainties, during which reconstruction uncertainties determine the magnitude
and variation of the total uncertainties.

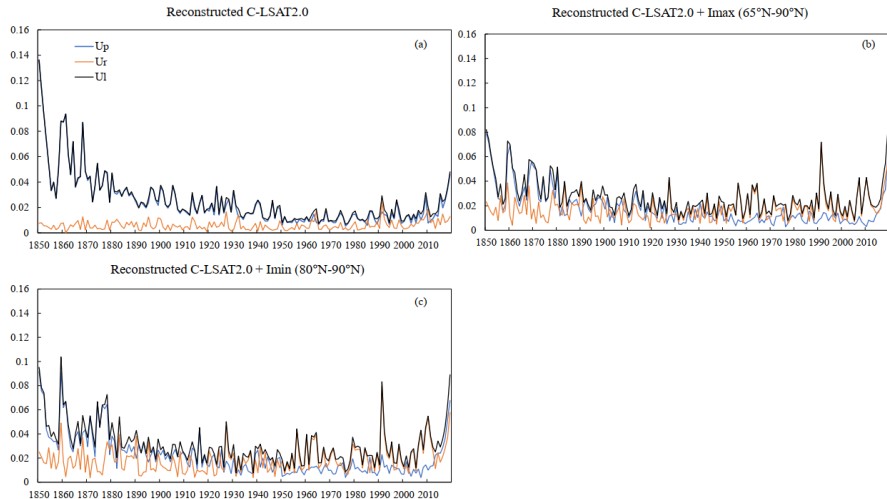

Figure 7 Parameter Uncertainty, reconstruction uncertainty and total uncertainty of three
reconstructed C-LSAT2.0 ensemble
**3.4.4 Uncertainty of global surface temperature**

The uncertainty of the global surface temperature consists of two components, the ocean
component and the land component, and we calculate the total global temperature uncertainty as the
sum of the two, based on the sea-to-land ratio, with the following formula:

$$U_g^2 = a \times U_l^2 + b \times U_s^2 \qquad (5)$$

where $U_g$ represents the total uncertainty of GMST, $U_l$ represents the uncertainty of global
averaged LSAT, here chosen from the reconstructed C-LSAT2.0; $U_s$ represents the uncertainty of
global averaged ocean component, here chosen from the ERSSTv5, since the uncertainty of
ERSSTv5 is only calculated up to 1854, our uncertainty of GST forward also only covers up to 1854.
a and b are constants, which are the proportion of land and ocean area to the globe, respectively, but
since the uncertainty of reconstructed Arctic region in CMST2.0-Imax and CMST2.0-Imin is
calculated according to the land uncertainty, a=0.32 and b=0.689 in CMST2.0-Imax and a=0.30 and
b=0.70 in CMST2.0-Imin.

Figure 8 shows uncertainties of the GMST, land component, and ocean component for CMST-
Interim (a), CMST2.0-Imax (b) and CMST2.0-Imin (c). The variation in GMST uncertainty is
similar for the three datasets, but the interannual variation in GMST uncertainty for CMST2.0-Imax
and CMST2.0-Imin is larger than CMST-Interim, especially after 1994, when both the magnitude
and interannual variation in GMST uncertainty for CMST2.0-Imax and CMST2.0-Imin are
significantly greater than CMST- Interim (Figure 8d). Uncertainties in the ocean and land
components have generally declined, and thus the uncertainty of GMST has also reduced (Figure
8a-c). Before 1870, the uncertainties of land and ocean component are similar, but the interannual
variability of the land uncertainty is greater than that of the ocean. During 1871-1986, the
uncertainty in the ocean component is larger than the uncertainty in the land component, and the
uncertainty of GMST depended mainly on the uncertainty in the ocean component, and the
interannual variability was consistent with the ocean component. There are two peaks in global
uncertainty during this period, in the late 1910s and early 1940s, consistent with ocean uncertainty.
The peaks in ocean uncertainty are associated with the two world wars, and the uncertainty is larger
due to the smaller observation coverage of the SST during the war period(Huang et al., 2020).
Between 1986 and 2003, the uncertainty of GST was determined by both the land and ocean
components. After 2003, the magnitude of uncertainty of the ocean component is smaller than that
of the land component, and the land component determines the magnitude of the uncertainty of GST,
and the interannual variation is also consistent with the land component.

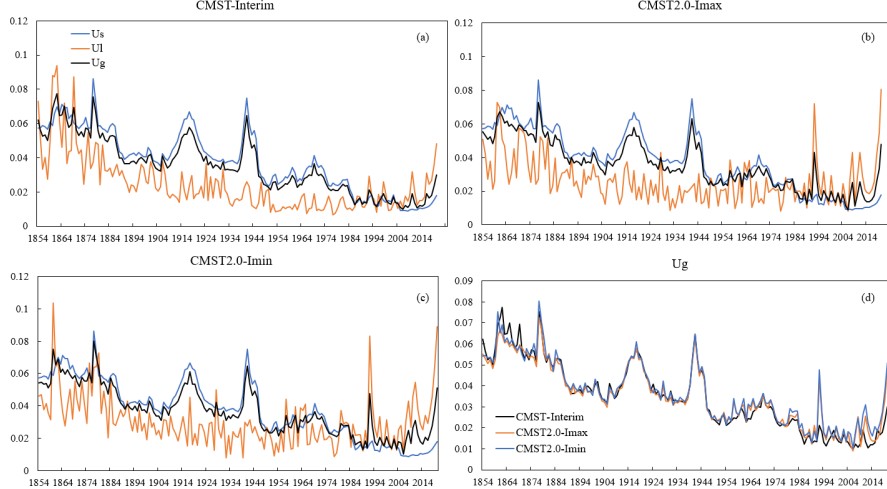

Figure 8 Uncertainties of GMST (Ug), LSAT (Ul) and SST (Us) for CMST-Interim (a), CMST2.0-
Imax (b) and CMST2.0-Imin (c) and their comparison of Ug(d).
**4.  Composition of C-LSAT2.0 and CMST2.0**
The C-LSAT2.0 datasets consist of two datasets, C-LSAT2.0 and reconstructed C-LSAT2.0, while
each dataset includes three temperature-related elements, including monthly average, maximum,
and minimum temperatures.
The CMST2.0 datasets consist of three versions: CMST2.0-Nrec, CMST2.0-Imax, and
CMST2.0-Imin.
CMST2.0-Nrec is the observation-based homogenized gridded dataset, consisting of C-
LSAT2.0 and ERSSTv5, where the uncertainty of C-LSAT2.0 is not estimated, and the uncertainty
of ERSSTv5 consists of parameter uncertainty and reconstruction uncertainty.
CMST2.0-Imax is based on CMST-Interim gridded dataset with the addition of Arctic
reconstruction (65N-90N), including reconstructed C-LSAT2.0 with the addition of Arctic





reconstruction (65N-90N) and ERSSTv5 with 90S-60N. Its uncertainties include the terrestrial
uncertainty and the oceanic uncertainty, where the terrestrial uncertainty is the uncertainty of the
reconstructed C-LSAT2.0 and of the reconstructed SAT over ice surface, including the parameter
uncertainty and the reconstruction uncertainty, and the oceanic uncertainty is derived from the
uncertainty of ERSSTv5 (Huang et al., 2017).
Similarly, CMST2.0-Imin is the gridded data, which modifies the reconstructed Arctic region
based on CMST2.0-Imin. The modification part is to reduce the reconstructed Arctic region of C-
LSAT2.0 to 80N-90N and expand the merged ERSSTv5 to 90S-80N area.
Table 2 Composition of CMST2.0 datasets and CMST-Interim.

| Versions | Timespan | LSAT | | SST | |
|---|---|---|---|---|---|
| | | datasets | uncertainty | datasets | uncertainty |
| **CMST2.0-Nrec** | 1850-2020 | C-LSAT2.0 | —— | ERSSTv5 | Parameter uncertainty + Reconstruction uncertainty |
| **CMST-Interim** | 1850-2020 | Reconstructed C-LSAT2.0 | Parameter uncertainty + Reconstruction uncertainty | ERSSTv5 | |
| **CMST2.0-Imax** | 1850-2020 | Reconstructed C-LSAT2.0 added Arctic reconstruction (65N-90N) | | ERSSTv5 (90S-65N) | |
| **CMST2.0-Imin** | 1850-2020 | Reconstructed C-LSAT2.0 added Arctic reconstruction (80N-90N) | | ERSSTv5 (90S-80N) | —— |

**5. The GMST series of CMST2.0 datasets**

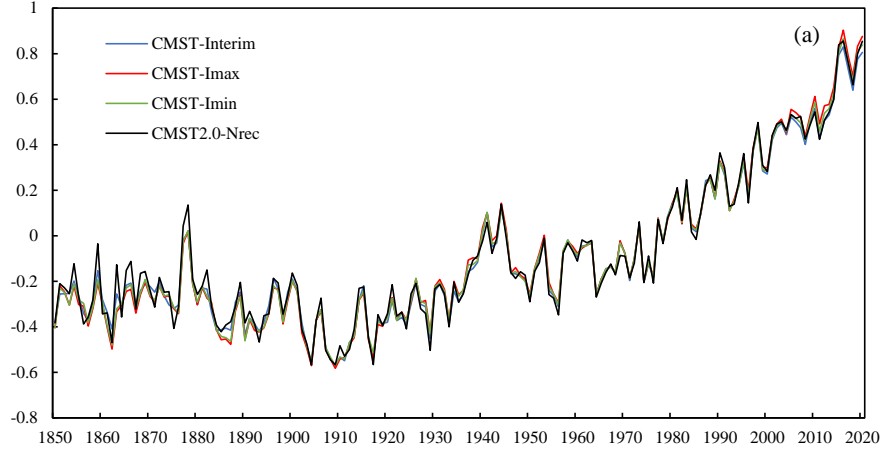


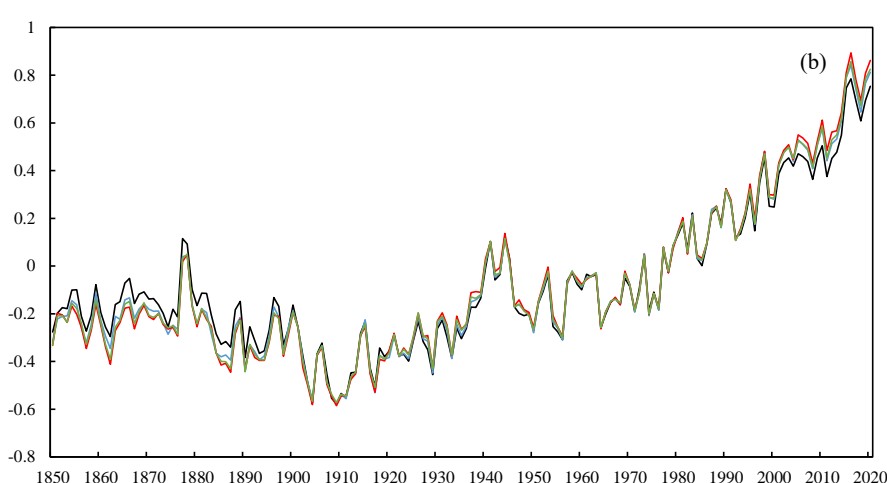


Figure 9 Comparison of GMST series for CMST2.0 datasets and CMST-Interim using two
methods: a) the mean of global mean LSAT and SST weighted the proportion of land and sea.; b)
calculated based on latitudinal weighting

Comparing the GMST series of CMST2.0 datasets and CMST-Interim shows that the variability
of GMST in the reconstructed datasets is generally consistent with CMST2.0-Nrec (Figure 9). We
also compare the GMST series for the four datasets calculated by the two methods, which is similar
for the three reconstructed datasets (CMST-Interim, CMST2.0-Imax and CMST2.0-Imin) and differ
slightly for the unreconstructed dataset CMST2.0-Nrec (Figure 9a & 9b). The warming of CMST-
Nrec in Figure 9b is significantly lower than that that in Figure 9a and , which is related to the lower
land coverage. The LSAT coverage of CMST2.0-Nrec was low in previous decades, which is below
18% before 1900 (Fig. 3), so the GMST series is susceptible to the influence of ocean temperature,
making the GMST series high; The LSAT coverage of CMST2.0-Nrec has increased in recent
decades, with terrestrial coverage above 70% (Figure 3), but the coverage is low at high latitudes,
in South America and Africa, where the absence of LSAT, especially at high latitudes and in the
Arctic, makes the GMST series low. It can be seen that the warming rate of CMST2.0-Nrec





calculated using latitude-weighting will be significantly lower, so we are using the sea-land ratio
method to calculate the warming trend when comparing each dataset in the following.
In Figure 9a, the CMST-Interim, CMST2.0-Imax and CMST2.0-Imin GMST series are lower
than CMST-Nrec before the 1880s, which is mainly due to the lower coverage of observations in
this period, making the interannual variability of the GMST series in CMST2.0-Nrec larger, while
the reconstructed datasets filled in part of the default grids, resulting in higher coverage and thus
lower interannual variability of GMST series. The reconstructed datasets show high agreement with
the CMST-Nrec temperature series and its interannual variability as the coverage of the observations
increased after the 1880s. While the GMST series of CMST2.0-Imax is significantly higher than the
other three datasets after the 2000s because CMST2.0-Imax reconstructs the Arctic region and the
polar amplification effect of the Arctic significantly increases the GMST series, the GMST series of
CMST-Interim and CMST2. 0-Imin are essentially the same as CMST-Nrec, but CMST2.0-Imin is
slightly higher than CMST-Interim because CMST2.0-Imin fills the 80N-90N region with ice
surface temperatures, while CMST-Interim uses SST. The GMST series of CMST2.0-Imax and
CMST2.0-Imin are higher than CMST-Interim after 2000, indicating that the influence of polar
temperature on global temperature also increases with global warming. In summary, the warming
trends of the reconstructed datasets for 1850-2020 are all higher than CMST2.0-Nrec
(0.05±0.003°C/10a), with CMST2.0-Imax having the most significant warming trend
(0.054±0.003°C/10a) and CMSR2.0-Imin the second largest (0.053±0.003°C/10a) (Table 3). The
warming trend estimated by CMST-Interim is 0.051 ± 0.003°C/10a, which is slightly larger than
CMST-Nrec, mainly due to the lower temperature series before the 1880s, excluding this period, the
warming trend from 1880 to 2020 estimated by CMST-Interim (0.073 ± 0.003°C/10a) is consistent
with CMST-Nrec (0.073 ± 0.004°C/10a) (Table). While the warming trends of CMST2.0-Imax and
CMST2.0-Imin are higher than the previous two datasets, 0.076±0.004°C/10a and
0.074±0.003°C/10a (Table 3), respectively, due to the polar amplification effect.
## 6. Comparison of CMST2.0-Imax and CMST2.0-Imin with other datasets

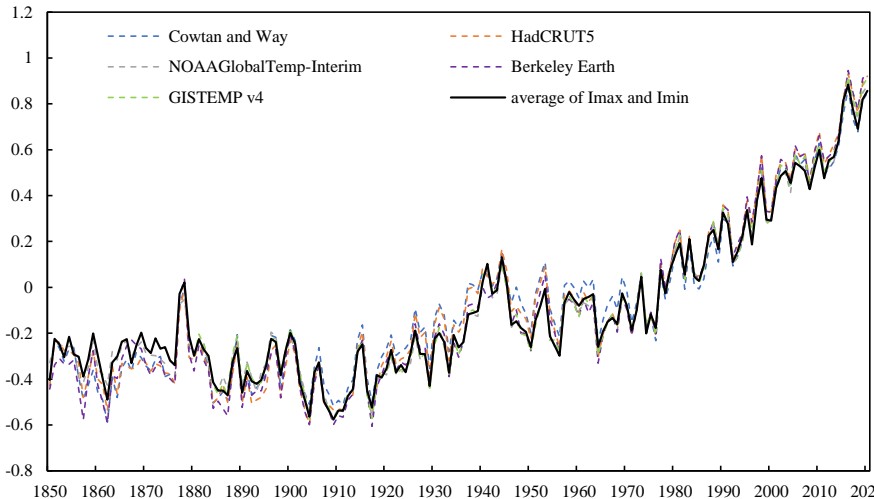


Figure 10 Comparison of GMST series for different datasets. The GMST series is the mean of
global mean LSAT and SST weighted the proportion of land and sea. The average of Imax and



Imin is the average of GMST series of CMST2.0-Imax and CMST2.0-Imin.

Figure 10 shows the GMST series of CMST2.0-Imax and CMST2.0-Imin compared with the
other datasets. The GMST series of the seven datasets are generally consistent. The GMST series of
CMST2.0-Imax and CMST2.0-Imin are similar to the other five datasets, indicating that their
estimated Arctic temperature variation is consistent with the other datasets, and can accurately
reflect the impact of the Arctic amplification effect on GST. Due to sparse observations, the
variability between datasets is high until the 1880s, as is the interannual variability between datasets.
After the 1900s, the GMST series of CMST2.0-Imax and CMST2.0-Imin are generally lower than
other datasets. In the 1910s-1970s, the Cowtan-Way dataset is consistently higher than other datasets.
In the 1930s-1950s, HadCRUT5 is higher than the other datasets, but similar to Cowtan-Way. After
the 2000s, the CMST2.0 datasets are generally lower than other datasets, with CMST2.0-Imax being
closer to the NOAAglobalTemp-Interim GMST series. For the period 1850-2020, the warming trend
of CMST2.0-Nrec is the lowest (0.05±0.003°C/10a) and the highest (0.062±0.003°C/10a) warming
trend is Berkeley in the seven datasets. The warming trend of CMST-Interim is consistent with
HadCRUT5, both at 0.051±0.003°C/10a. The warming trend of CMST2.0-Imax is the same as
NOAAglobalTemp-Interim (0.054±0.003°C/10a). Between 1880 and 2020, CMST2.0-Nrec
(0.073±0.004°C/10a) is agreement with CMST-Interim (0. 073±0.003°C/10a), CMST2.0-Imax is
consistent with NOAAglobalTemp-Interim (0.076±0.004°C/10a), and CMST2.0-Imin
(0.075±0.003°C/10a) is consistent with Cowtan -Way (0.074±0.003°C/10a) (Table 3). We also
calculate the warming trends of different datasets for different periods 1900-2020, 1951-2020, 1979-
2020 and 1998-2020 and found that the warming rate becomes faster over time for most of the
datasets, especially the increasing warming trend for 1998-2020 is much larger than the other
periods, indicating that the global warming rate is accelerating. The maximum warming trend of
0.228±0.029°C/10a (GISTEMP v4) during 1998-2020 increased by 0.037±0.017°C/10a compared
to the warming trend during 1979-2020. the largest increasing warming trend is NOAAglobalTemp-
Interim, with a warming trend of 0. 037 ± 0.017°C/10a for 1998-2020, which is 0.04°C/10a higher
than the warming trend during 1979-2020, followed by CMST2.0-Imax, CMST2.0-Imin and
Berkeley Earth, CMST2.0-Nrec and CMST-Interim have relatively small increases in the warming
trend. The relatively large increases of warming trend estimated in most datasets with reconstructed
Arctic temperatures, compared to those without (CMST2.0-Nrec and CMST-Interim), illustrate the
impact of polar amplification on global warming and reflect the importance of reconstructing Arctic
default data.

Table 3 Warming trends for different datasets during different periods. The GMST series used to
calculate the warming trend is the mean of global mean LSAT and SST weighted the proportion of
land and sea.

| | CMST2.0 -Nrec | CMST-Interim | CMST-Imax | CMST-Imin | Cowtan − Way | HadCRUT5 | NOAAglobal Temp-Interim | Berkeley Earth | GISTEMP v4 |
|---|---|---|---|---|---|---|---|---|---|
| **1850-2020** | 0.050±0.003 | 0.051±0.003 | 0.054±0.003 | 0.053±0.003 | 0.058±0.003 | 0.051±0.003 | 0.054±0.003 | 0.062±0.003 | — |
| **1880-2020** | 0.073±0.004 | 0.073±0.003 | 0.076±0.004 | 0.075±0.003 | 0.074±0.003 | 0.081±0.004 | 0.076±0.004 | 0.083±0.004 | 0.077±0.004 |
| **1900-2020** | 0.091±0.004 | 0.090±0.004 | 0.093±0.004 | 0.091±0.004 | 0.084±0.004 | 0.094±0.004 | 0.093±0.004 | 0.099±0.004 | 0.095±0.004 |
| **1951-2020** | 0.145±0.007 | 0.139±0.007 | 0.146±0.007 | 0.143±0.007 | 0.130±0.008 | 0.150±0.008 | 0.147±0.007 | 0.155±0.008 | 0.151±0.007 |
| **1979-2020** | 0.174±0.013 | 0.168±0.011 | 0.184±0.011 | 0.179±0.011 | 0.190±0.012 | 0.193±0.012 | 0.184±0.012 | 0.195±0.012 | 0.191±0.012 |
| **1998-2020** | 0.198±0.030 | 0.19±0.027 | 0.212±0.026 | 0.209±0.026 | 0.189±0.028 | 0.215±0.028 | 0.224±0.028 | 0.220±0.030 | 0.228±0.029 |


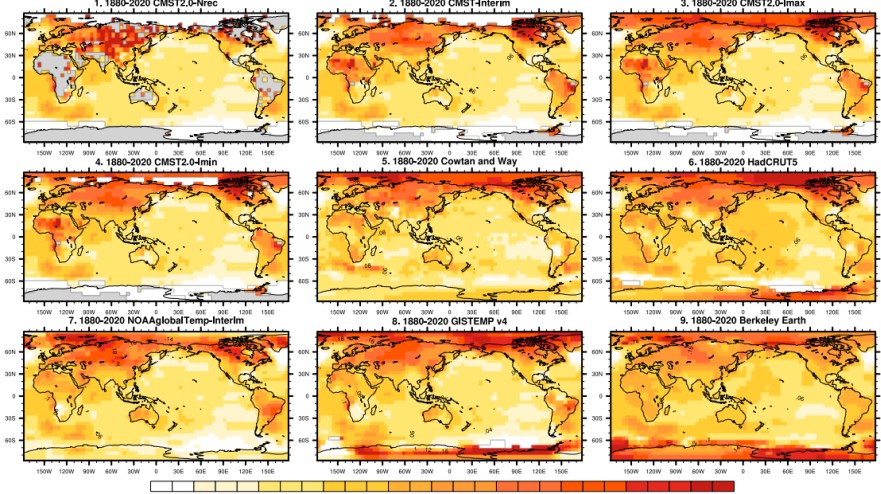

Figure 11 Distribution of warming trends estimated from different datasets during 1880-2020.
Figure 11 compares the distribution of warming trends for different datasets for 1880-2020.
The distribution of warming trends is relatively consistent among the nine datasets except for the
Antarctic, with a zone of high warming values in central Asia and Europe, and northeastern North
America. There are large differences among the datasets in the Antarctic region due to the sparse
observations. CMST-Interim, CMSR2.0-Imax and CMST2.0-Min have fewer LSATs in the
Antarctic due to the sparse observations and observational constraints. Except for CMST2.0-Nrec,
the estimated warming trends of the other eight datasets have a clear trend of increasing with latitude
in the Northern Hemisphere region. Most datasets assess a significantly higher warming trend in the



Arctic (60N-90N) than in the lower latitudes. Except for the CMST2.0-Nrec and CMST-Interim
datasets in which Arctic temperature is not available, the magnitude of the estimated Arctic warming
trend for 1880-2020 is similar. Still, the warming trends near the poles differ significantly, with more
significant warming trends estimated by HadCRUT5 and GISTEMP v4. CMST2.0-Imax,
CMST2.0- Imin, Cowtan-Way and Berkeley Earth have similar warming trends, while
NOAAglobalTemp-Interim has the smallest warming estimate near the poles. CMST2.0-Imax,
HadCRUT5, and GISTEMP v4 all show a high warming trend in the high latitudes of North America
and the northwestern Arctic Ocean, but CMST2.0-Imax has a relatively small range of highs.
Cowtan-Way and Berkeley Earth are similar to the former three datasets, but with smaller ranges
and magnitudes. Meanwhile, each dataset also has a range of warming highs in the southeastern
Arctic Ocean, NOAAglobalTemp-Interim estimates the most extensive range of warming,
CMST2.0-Imax, CMST2.0-Min, HadCRUT5, and GISTEMP v4 estimate similar ranges of
warming. In addition, all datasets, including CMST2.0-Nrec and CMST-Interim, have low warming
trend near Scandinavia. The analysis of the warming trends in the Arctic shows that the magnitude
and spatial distribution of the warming trends estimated based on CMST2.0-Imax and CMST-Imin
are more consistent with the other datasets. Therefore they are reasonable for the spatial
interpolation reconstruction of temperature anomalies in the Arctic.
**7.  Summary and Prospects**
This paper describes the composition and construction process of the latest versions of the C-
LSAT 2.0 and CMST 2.0 ensemble datasets. The C-LSAT 2.0 datasets consist of the C-LSAT 2.0
gridded dataset and the reconstructed C-LSAT 2.0 dataset, including three meteorological elements:
monthly average, maximum and minimum temperatures. The CMST2.0 datasets consist of the
CMST 2.0-Nrec gridded dataset and two reconstructed datasets (including CMST 2.0-Imax and
CMST2.0-Imin). The CMST 2.0 datasets contain the monthly average temperature anomaly. The
resolution of all datasets is 5x5 and the time range is 1850-2020. The reconstructed C-LSAT 2.0
dataset, reconstructed according to the high- and low-frequency reconstruction method in Sun et al.
(2021), is merged with ERSSTv5 to generate the global surface temperature ensemble dataset
CMST-Interim. CMST 2.0-Imax and CMST 2.0-Imin are based on CMST-Interim, combining IDW
and high- and low-frequency reconstruction methods for temperature reconstruction in the Arctic.
Compared with the unreconstructed dataset CMST 2.0-Nrec, the coverage of the reconstructed
datasets is greatly improved. These two datasets have greatly improved coverage in the Northern
Hemisphere due to the reconstruction in the Arctic. Compared to 60%-70% for CMST 2.0-Nrec
before 1910, the coverage of CMST-Interim has improved to 75%-85%, and CMST 2.0-Imax and
CMST 2.0-Imin are both above 80%. The coverage of CMST 2.0-Imax and CMST2.0-Imin in the
Northern Hemisphere is 80%-99% and CMST-Interim is 65%-87%. In the Southern Hemisphere,
there was no difference in coverage between the three reconstructed datasets.
We then systematically evaluate the uncertainty of the reconstructed datasets. The results of
the uncertainty assessment of the reconstructed C-LSAT 2.0 show that the magnitude of the
reconstruction uncertainty is generally smaller than that of the parameter uncertainty, and the
parameter uncertainty mainly determines the total uncertainty of the LSAT. The uncertainty of the
reconstructed LSAT is similar to previous estimates (Li et al., 2020; Sun et al., 2021). The
uncertainty of reconstructed C-LSAT2.0+Imax and reconstructed C-LSAT2.0+Imin is relatively
consistent with the uncertainty variation of reconstructed C-LSAT2.0, but the interannual variation
is larger, and the increasing trend of parameter uncertainty of reconstructed C-LSAT2.0+Imax and



reconstructed C-LSAT2.0+Imin is significantly higher than that of reconstructed C-LSAT2.0 after
2017. The uncertainty analysis of CMST 2.0 shows that the uncertainty of GST depends mainly on
the oceanic component before 1986, is determined by both oceanic and terrestrial components
during 1986-2003, and depends on the magnitude of the terrestrial component after 2003.
Results comparing the GMST series of the three CMST 2.0 datasets and CMST-Interim show
that the reconstructed datasets improve the estimation of global warming trends while increasing
data coverage, especially for the datasets that include the Arctic region in the reconstructed area.
Compared with 0.05 ±0.003°C/10a and 0.073 ±0.004°C/10a for CMST 2.0-Nrec, CMST 2.0-Imax
and CMST 2.0-Imin estimated warming trends of 0.054 ±0.003°C/10a and 0.053 ±0.003°C/10a for
1850 -2020 and 1880 -2020 is 0.076 ±0.004°C/10a and 0.075 ±0.003°C/10a, with a very significant
increase. Compared with the five datasets in IPCC AR6, it can be found that the datasets considering
the reconstruction of Arctic sea ice temperature can more accurately reflect the effect of polar
amplification on global temperature, and the GMST series and warming trends estimated by CMST
2.0-Imax and CMST 2.0-Imin are more consistent with these five datasets, and both have similar
estimates of the spatial distribution and magnitude of warming trends in the Arctic as the other
datasets.
The current CMST 2.0 dataset for the Arctic is a reconstruction of the sea ice surface
temperature in a defined region (65°N-90°N or 80°N-90°N) with 2 meters air temperature. Although
the influence of Arctic temperature on global temperature is considered and the change of GMST
series is estimated relatively accurately, it still cannot reflect the impact of sea ice dynamics on
global temperature very accurately. Therefore, our future work will gradually consider the dynamics
of sea ice as much as possible in the reconstruction process in order to more accurately estimate and
analyze the amplification effect of the Arctic and its impact on GMST.
Last but not the least, due to the limited observations, it is very difficult to fully reconstruct the
SATs over the Antarctic and the surrounding SSTs during the earlier periods (for example: prior to
1950s), which made the CMST2.0 is still not "fully" coverage. This will need to be better addressed
by continuing to supplement data sources and refining technical methods in future studies.

## 8. Data availability

The C-LSAT2.0 datasets are currently publicly available at the website of figshare under the DOI
https://doi.org/10.6084/m9.figshare.16968334.v4 (Sun and Li, 2021b), which contains monthly
mean, maximum and minimum temperature before and after reconstruction during 1850-2020. The
CMST2.0 datasets can be downloaded at https://doi.org/10.6084/m9.figshare.16929427.v4 (Sun
and Li, 2021a), which contains CMST2.0-Nrec, CMST-Interim, CMST2.0-Imax and CMST2.0-
Imin datasets.
**Author contributions.** All co-authors were involved in data collection, data analysis, and dataset
development. QL was primarily responsible for writing the paper and constructing the dataset. QL
and WS conceived the study design with the participation of all co-authors. All authors were
involved in the writing of the paper.
**Competing interests.** The authors declare that they have no conflict of interest.
**Acknowledgments.** This study is supported by the Natural Science Foundation of China (Grant:





41975105), the National Key R&D Program of China (Grant: 2018YFC1507705;
2017YFC1502301).

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
