# Peer review of "Description of the China global Merged Surface Temperature version 2.0"

_Earth System Science Data, 2021_

## Author Response (AR1)

Reviewer(s)' Comments to Author:

Reviewer: 1

Comments to the Author

The study takes great effort to complement the well known global database. The following suggestions aim to improve the ms to convince more potential users:

Tables would facilitate reading than the detail text;

List a table to compare the major global databases: input & output data, mechanism; merits and limitations;

Assess the representative of the data in years and in areas. A sample would be biased to a population if the population is spatial stratified heterogeneity and not all strata of the population are covered by the sample.

Reviewer: 2

Comments to the Author

In this study, an updated global merged surface temperature dataset was generated and described, with data coverage improved to even cover high latitudes in the polar regions after reconstruction. Overall, the paper is well organized and the technical flow is clearly delineated. Therefore, i recommend its acceptance provided the authors address the following minor comments.

Line 52: it could be "How to account for this deficiency…", deficiency should not be improved.

Line 70: '…, which was generated by merging China…'

Line 154: references are need here to describe the proposed "IDW extrapolation method".

Section 3.2.1: how to deal with data in the boundary when performing running means with a given window? Padding data at the boundary or simply using the available data in within the window?

Table 2: spatial and temporal resolutions of each dataset could be provided.

Table 3: the warming trend appears to be systematically underestimated by CMST-2.0 when compared with other datasets, any specific reasons?

Figure 11: a trend deviation map helps better interpret difference between different GMST datasets.

**Response to Reviewers and Editor**

We gratefully thank the editor and all reviewers for their time spent making their constructive remarks and valuable suggestions, which has significantly raised the quality of the manuscript and has enabled us to improve the manuscript. Below the comments of the reviewers are response point by point and the revision are indicated.

Reviewer: 1

Comments to the Author

The study takes great effort to complement the well known global database. The following suggestions aim to improve the ms to convince more potential users:

Tables would facilitate reading than the detail text;

List a table to compare the major global databases: input & output data, mechanism; merits and limitations;

Reply: We have added new Table 3 in the manuscript to introduce the general information of each dataset. As follows:

Table 1 General information of input datasets

|  | Period of record | Land component | SST component | resolution | Interpolation, reconstruction, and uncertainties evaluation |
|---|---|---|---|---|---|
| **China-MST2.0** | 1850-2020 | China-LSAT2.0 | ERSSTv5 | 5°×5° | Spatial smoothing and EOTs; observational constraint; ensemble uncertainties |
| **HadCRUT5** | 1850-2020 | CRUTEM5 | HadSST4 | 5°×5° | Gaussian process method; observational constraint; ensemble uncertainties |
| **NOAAGlobal-Interim** | 1850-2020 | GHCNv4 | ERSSTv5 | 5°×5° | Spatial smoothing and EOTs; ensemble uncertainties |
| **GISTEMP v4** | 1880-2020 | GHCNv4 | ERSSTv5 | 2°×2° | Spatial interpolation methods over reasonable distances; ensemble uncertainties |
| **Berkeley Earth** | 1850-2020 | Berkeley | HadSST4 | 1°×1° | Kriging-based spatial interpolation with constant distance parameters at all latitudes |
| **Cowtan and Way** | 1850-2020 | CRUTEM4 | HadSST3 | 5°×5° | Kriging-based method with constant distance parameters at all latitudes |

Assess the representative of the data in years and in areas. A sample would be biased to a population if the population is spatial stratified heterogeneity and not all strata of the population are covered by the sample.

Reply: Thanks for the comment.

You are right that a sample would be biased to a population if the population is spatial stratified

heterogeneity and not all strata of the population are covered by the sample. However, the existing experiment study shows that this kind of bias should be limited and minor for the global surface temperature dataset in different years and regions, the warming trends are similar with the estimation with commonly used methods (Li et al., 2017; Wang et al., 2014; 2017).

But your comments deserve consideration, more complicated methods will have the advantage in estimation appearing in the areas with few stations and in the early years, when stations had sparse coverage and were unevenly distributed, or for other variables like precipitation, etc.

Ref:

Li Q., Zhang L., Xu W., et al., 2017, Comparisons of time series of annual mean surface air temperature for China since the 1900s: Observation, Model simulation and extended reanalysis. *Bull. Amer. Meteor. Soc.*, 98(4): 699 − 711, doi: 10.1175/BAMS-D-16-0092.1 (IF=8.766)

Wang J., Xu C., Hu M.,et al, 2014, A New Estimate of the China Temperature Anomaly Series and Uncertainty assessment in 1900-2006, *J. Geophys. Res. Atmos.*, 119 (1-9), doi:10.1002/2013JD020542.

Wang J., Xu C., Hu M , et al, 2017, Global land surface air temperature dynamics since 1880, *International Journal of Climatology*, 38: e466-e474, DOI: 10.1002/joc.5384

Reviewer: 2

Comments to the Author

In this study, an updated global merged surface temperature dataset was generated and described, with data coverage improved to even cover high latitudes in the polar regions after reconstruction. Overall, the paper is well organized and the technical flow is clearly delineated. Therefore, i recommend its acceptance provided the authors address the following minor comments.

Line 52: it could be "How to account for this deficiency…", deficiency should not be improved.
Reply: done.

Line 70: '…, which was generated by merging China…'
Reply: done.

Line 154: references are need here to describe the proposed "IDW extrapolation method".
Reply: Thanks.

It should be Adjusted IDW used in Cheng et al (2020), we have cited the relevant literature in revised manuscript when the AIDW was firstly referred in line 140.

Section 3.2.1: how to deal with data in the boundary when performing running means with a given window? Padding data at the boundary or simply using the available data in within the window?
Reply: Thanks.

We use only the data available in the window when performing sliding averaging in the longitude direction. In the latitudinal direction, the longitude is different from the latitude and there is no boundary effect.

Table 2: spatial and temporal resolutions of each dataset could be provided.

Reply: All the versions of CMST should be in the resolution of 5°×5° in the latitudal and longitudal directions. Per your request, we clarified this in the section 1 (lines 77-78).

Table 2 General information of input datasets

| | Period of record | Land component | SST component | resolution | Interpolation, reconstruction, and uncertainties evaluation |
|---|---|---|---|---|---|
| **China-MST2.0** | 1850-2020 | China-LSAT2.0 | ERSSTv5 | 5°×5° | Spatial smoothing and EOTs; observational constraint; ensemble uncertainties |
| **HadCRUT5** | 1850-2020 | CRUTEM5 | HadSST4 | 5°×5° | Gaussian process method; observational constraint; ensemble uncertainties |
| **NOAAGlobal-Interim** | 1850-2020 | GHCNv4 | ERSSTv5 | 5°×5° | Spatial smoothing and EOTs; ensemble uncertainties |
| **GISTEMP v4** | 1880-2020 | GHCNv4 | ERSSTv5 | 2°×2° | Spatial interpolation methods over reasonable distances; ensemble uncertainties |
| **Berkeley Earth** | 1850-2020 | Berkeley | HadSST4 | 1°×1° | Kriging-based spatial interpolation with constant distance parameters at all latitudes |
| **Cowtan and Way** | 1850-2020 | CRUTEM4 | HadSST3 | 5°×5° | Kriging-based method with constant distance parameters at all latitudes |

Table 3: the warming trend appears to be systematically underestimated by CMST-2.0 when compared with other datasets, any specific reasons?

Reply: Thanks.

First of all, due to the data resources and processing methods, the warming trends estimated from different datasets would have some differences. For example, from 1880 through 2020, the warming trends estimated from CMST2.0-Imax and CMST2.0-Imin are broadly consistent with those from NOAAGlobal-Interim and GISTEMP v4, lower than those from HadCRUT5 and Berkeley Earth, and higher than those from Cowtan and Way (Table 4 in the revised manuscript). All of the datasets have been used to contribute to the evaluation of the GMST warming trends in the newly released IPCC AR6 (Gulev et al., 2021).

Secondly, the differences among all the datasets are not statistically significant (their estimated warming trend ranges are largely overlapping when the uncertainties of the linear trend have been considered). Table 4 compares the global surface temperature trends for different datasets at different periods. In addition, since we focus on the reconstruction scenarios in the polar regions of the CMST in this paper, we also compare the distribution of temperature trends in the Arctic for different datasets in lines 494-510.

In summary, although there are some differences, the warming trends estimated by CMST2.0-

Imax and CMST2.0-Imin in CMST2.0 do not systematically underestimate the warming trends if we consider the warming trends with the uncertainty range.

Ref:

Gulev, S. K., P. W. Thorne, J. Ahn, F. J. Dentener, C. M. Domingues, S. Gerland, D. Gong, D. S. Kaufman, H. C. Nnamchi, J. Quaas, J. A. Rivera, S. Sathyendranath, S. L. Smith, B. Trewin, K. von Shuckmann, R. S. Vose, 2021, Changing State of the Climate System In: Climate Change 2021: The Physical Science Basis. Contribution of Working Group I to the Sixth Assessment Report of the Intergovernmental Panel on Climate Change [Masson-Delmotte, V., P. Zhai, A. Pirani, S. L. Connors, C. Péan, S. Berger, N. Caud, Y. Chen, L. Goldfarb, M. I. Gomis, M. Huang, K. Leitzell, E. Lonnoy, J. B. R. Matthews, T. K. Maycock, T. Waterfield, O. Yelekç, R. Yu and B. Zhou (eds.)]. Cambridge University Press. In Press.

Figure 11: a trend deviation map helps better interpret difference between different GMST datasets.

Reply: Thanks.

   Since we cannot take any dataset as truth of the observation, we added Figure 12 in the manuscript, which shows the distribution of differences in warming trends estimated with CMST2.0-Imax for different datasets.

[Figure]

Figure 12 Differences in warming trends estimated by other 6 datasets (including CMST2.0-Imin) and CMST2.0-Imax